# Robotic Surgery for Elective Repair of Visceral and Renal Artery Aneurysms: A Systematic Review

**DOI:** 10.3390/jcm13123385

**Published:** 2024-06-09

**Authors:** Luigi Federico Rinaldi, Chiara Brioschi, Enrico Maria Marone

**Affiliations:** 1Vascular Surgery, Department of Integrated Surgical and Diagnostic Sciences, University of Genoa, 16132 Genoa, Italy; 2Hospital Policlinico di Monza, 20900 Monza, Italy; chiara.brioschi@policlinicodimonza.it (C.B.); enricomaria.marone@gmail.com (E.M.M.); 3Vascular Surgery, Department of Clinical-Surgical, Diagnostic and Pediatric Sciences, University of Pavia, 27100 Pavia, Italy

**Keywords:** robotic surgery, visceral aneurysm repair, laparoscopic vascular surgery

## Abstract

**Background:** The treatment choice of visceral artery aneurysms in an elective setting is debated. The durability and the risk of reintervention with endovascular treatment are still reasons for concern, whereas open surgery is invasive and burdened by major complications. In anecdotal reports and isolated studies, robotic-assisted surgery seems to provide the possibility of a minimally invasive treatment and the durability of traditional open surgery, but the literature supporting this view is scarce. This review aims to collect the results of robotic-assisted surgery in the treatment of visceral artery aneurysms. **Methods:** A systematic search of the main research databases was performed: the study endpoints were mortality and conversion rates, perioperative morbidity, and freedom from late complications and reinterventions. **Results**: We identified 16 studies on 53 patients. All cases underwent successful resection, with three conversions to laparoscopy. Perioperative and aneurysm-related mortality were nil. Over a median follow-up of 9 months, two reinterventions were reported (3.6%). **Conclusion:** The robotic technique is safe and effective in treating splenic and renal artery aneurysms, and it should be considered as a valuable alternative to endovascular and open repair, although larger sample sizes and a longer-term follow-up are necessary to confirm such results.

## 1. Introduction

The emergence of robotic-assisted laparoscopic surgery (RALS) has changed the approach to many diseases in the field of abdominal surgery, urologic surgery, and orthopedics [1,2,3]. Although its application to vascular surgery is still limited to selected anecdotal cases, the treatment of visceral artery aneurysms (VAA) is a field where robotic surgery is most promising.

Visceral artery aneurysms (VAAs) and pseudoaneurysms (VAPAs), despite their rarity in the general population, have a high impact in terms of mortality and morbidity due to the risk of sudden rupture, which is particularly threatening in pregnant women, patients affected by hepato-biliary diseases (portal hypertension, pancreatitis, transplant recipients), and in some forms of vasculitis [4,5]. The indications and the treatment modalities of intact aneurysms are still controversial due to the paucity and heterogeneity of the available data and the complications of both open and endovascular repair [6].

Endovascular exclusion is now the most frequent therapy for intact aneurysms and pseudoaneurysms deemed at high risk of rupture, since it has lower perioperative mortality than open surgery, but it has important limitations in durability, along with the risk of end-organ ischemia (post-embolization syndrome or occlusion of the parent artery) [4,5].

The possible advantages of robotic surgery include a less invasive treatment than traditional open surgery and a more radical solution if compared with endovascular techniques.

This study aims to review the outcomes of robotic treatment of visceral and renal artery aneurysms (RAAs) reported in the contemporary literature.

## 2. Materials and Methods

A systematic search of PubMed/MEDLINE, EMBASE, and Web of Science (WOS) was performed by two independent authors (LFR and CB) using the terms “Robotic” or “Robotic-assisted surgery” and “visceral” or “splanchnic” or “mesenteric” or “renal” aneurysm.

The study selection process and data reporting were carried out in accordance with the PRISMA Guidelines (Figure 1) [7].

All the primary literature reporting on visceral or renal aneurysms or on pseudoaneurysms treated by RALS was included without restrictions of volume, language, or year of publication. The inclusion criteria required that baseline clinical data, operative technique, perioperative and postoperative outcomes were indicated. Studies not reporting on original clinical data and reviews were excluded.

We recorded the patients’ clinical characteristics, the aneurysm size and location, the indications to surgical treatment, and the reasons for having chosen robotic-assisted treatment over other options, if reported. Concerning the operative data, our review included all available information about the access technique, total operative time (OT) and cross-clamping time, and target vessel reconstruction. Regarding the perioperative and postoperative course, the search focused on mortality, the occurrence of systemic complications and major adverse events, open surgical conversions or reinterventions, length of in-hospital stay, and longer-term results at follow-up.

The primary study endpoints were technical success, mortality, and freedom from conversions. The secondary endpoints were perioperative morbidity, in-hospital stay, and freedom from late complications and reinterventions at follow-up. The results were compared with the outcomes of open, endovascular, and laparoscopic treatment of VAAs reported in the most recent primary studies.

Continuous variables are reported as median and interquartile range (IQR) or mean and standard deviation (SD). Categorical variables are expressed as numbers and percentages.

## 3. Results

### 3.1. Patients

The query identified 17 studies published between 2010 and 2022 reporting on robotic repair of VAAs in 54 patients (25 females, 9 males, 20 non-reported; median age: 56; interquartile range: 11) [8,9,10,11,12,13,14,15,16,17,18,19,20,21,22,23,24,25]. They were all true aneurysms; 30 of them involved the splenic artery, 22 the renal artery, 1 the hepatic artery, and 1 the coeliac trunk (Table 1). Aneurysm size ranged between 15 and 62 mm (IQR: 13).

All patients were treated in an elective setting for asymptomatic or symptomatic but intact VAAs. The reasons for choosing robotic-assisted surgery were provided in only 13 cases. The most frequent reason was the unsuitability for endovascular exclusion due to either extremely distal location (the splenic or renal hilum in eight cases, branch point of a renal artery in two cases), or non-specified unfavorable position and size of the aneurysm (two cases). Ceccarelli et al. justified the choice of robotic surgery with a high risk of post-embolization syndrome and splenectomy in a SAA but did not specify how such risk was estimated [13].

The robotic system most frequently employed was the DaVinci Surgical System^®^ (Intuitive Surgical, Sunnyvale, CA, USA); this system was used in 51 cases. One renal artery aneurysm (RAA) was treated via robotic-assisted coil embolization using the Magellan System^®^ (Auris Surgical Robotics, Redwood City, CA, USA) [23]. In four cases, the aid of 3D-printed models to plan and perform the intervention was described, and one case reported using mixed reality for intraoperative guidance [12,17,24].

### 3.2. Operative Technique

The operative techniques described for splenic and renal artery aneurysms showed few variations across the studies included.

For splenic aneurysm repair, the patient is placed in a supine position and five ports (two of 12 mm and three of 8 mm) are placed after inducing the pneumoperitoneum. The artery is accessed through the gastrocolic ligament and exposed on the upper pancreatic edge. After endovenous heparin administration, the smallest collateral vessels are ligated, and the main vessel is clamped upstream and downstream of the aneurysm. The aneurysm is excised and removed in an Endobag, and the artery is reconstructed via direct anastomosis (13 cases) or graft interposition with interrupted stitches or running sutures in Polypropylene 6/0 or 7/0 (2 cases, technique non-reported in 15 cases).

Regarding renal artery aneurysms, most authors used the transperitoneal approach: the patient is placed in semi-lateral decubitus at 30 degrees, and six ports are used. The left renal hilum is exposed through a laterocolic approach, mobilizing the left colonic flexure and dividing the splenocolic ligament. On the right, the kidney is accessed by mobilizing and retracting medially the duodenal loop and the head of the pancreas. After exposing the aneurysm, all the collateral branches are skeletonized, and the aneurysm is resected [9,10,13,14,15,18,19,21,24]. Some authors use ice-cold Ringer’s lactate selective perfusion if a prolonged time of renal ischemia is expected [14]. The reconstruction techniques included aneurysm excision with patch repair in 8 cases, direct anastomosis with PTFE CV 6 or CV7 running sutures in 11 cases, prosthetic graft interposition (1 case), or autologous bench-prepared Y-shaped saphenous-vein grafts (2 cases). Wu et al. adopted a retroperitoneal approach, reporting shorter operative time and less blood loss than the transperitoneal access [11].

Robotic surgery of hepatic artery and coeliac trunk aneurysms are described in one case each and required five ports (two 12-mm, two 8-mm, and one 5-mm); the patient was placed in a supine, 30 degrees reverse Trendelenburg position [16,17]. The target vessels were accessed through the gastrocolic ligament, and the aneurysms were resected with direct reconstruction or simple ligation [18,19].

### 3.3. Inclusion and Exclusion Criteria for Robotic Surgery

Although the current literature does not point out specific situations in which robotic surgery should be considered, selection criteria for RALS were specified in 13 cases: the most frequent reported reason for choosing robotic treatment was aneurysm location at the splenic or renal hilum or at a branch point of the parent artery (nine cases), followed by concerns due to the aneurysm size (two cases), generic unsuitability for endovascular treatment (one case), and unspecified high risk for target vessel ischemia (one case).

Very few authors reported on exclusion criteria, but Stadler and Giulianotti, who both proposed RALS as the first-choice treatment in fit patients with VAAs, excluded those affected by heavy cardiopulmonary comorbidities, contraindicating laparoscopic surgery [8,21]. In addition, Stadler et al. reported a hostile abdomen as exclusion criterion, whereas the other authors do not specify if patients with previous abdominal surgery could be or were treated with RALS [9,10].

### 3.4. Perioperative Results

Overall, 50 cases of RALS were successful, and three SAAs underwent intraoperative open conversion to laparoscopic or open surgery (one with concomitant splenectomy) due to flat adhesion of the splenic artery to the abdominal cavity. The operative time was reported in 13 studies (mean: 169.3′, SD: 83.3′), and the time of end-organ ischemia was reported in 11 (mean: 39.7′, SD: 10′). Surgery was significantly longer in RAA treatment (mean; 130′, SD: 50) than in SAAs (mean: 218′, SD: 66, SE difference: 18.239, *p*-value < 0.001), but SAAs entailed a slightly longer mean time of ischemia (45.5′ vs. 37.8′, SE difference: 1.930, *p*-value: 0.002).

Systemic and local postoperative complications are rarely reported and never severe: Ceccarelli et al. described a hematocrit drop on postoperative day II requiring blood transfusion after a RAA repair, but the mean reported intraoperative blood loss was extremely low for all kinds of VAAs (88 mL, SD: 53) [13]. Salloum et al. reported a peak transient in bilirubin and liver enzymes on postoperative day I after SAA repair, which returned to the normal range on the following day, without clinical implications [18]. A worsening of creatinine levels and renal function was reported in one case of RAA repair by Giulianotti et al., but it did not require hemodialysis nor specific treatment, and it resolved spontaneously [19].

Most studies report a quick recovery, with mobilization and liquid diet started on postoperative day I and solid diet on postoperative day II. The mean length of in-hospital stay was 4.2 days (SD:1.6), and it was significantly longer in patients treated for SAAs (SE: 0.324, *p*-value < 0.001) (see Table 2).

No perioperative deaths waere reported. Over a median follow-up period of 9 months (range: 2–48), no aneurysm-related deaths occurred. Three patients (5.7%) underwent reinterventions during the follow-up: two cases (3.8%) of splenic infarction were treated by partial robotic splenectomy, and one renal artery stenosis (1.9%) required percutaneous angioplasty. Notably, one case of splenic infarction was due to the unfeasibility to reconstruct the parent artery after aneurysm resection because of failure of the robotic system that caused a blockade of one of the mechanical arms, but an intraoperative open conversion had not been performed because the collateral network from the gastric vessels were considered able to provide an adequate compensation for the resected splenic artery [21].

## 4. Discussion

Laparoscopic and robotic treatments of VAAs are widely described in the literature and are subjects of interest [8,9,10,11,12,13,14,15,16,17,18,19,20,21,22,23,24]. However, both techniques have found limited employment in everyday clinical practice, mainly due to the lack of expertise among vascular surgeons and the cost of the facilities and maintenance of surgical robots; in addition, endovascular repair continues to represent the first-choice treatment in most cases [5].

That, along with the rarity of the disease, makes it extremely difficult to collect high-quality data based on large numbers. Evidence concerning RALS in this setting is extremely scarce, but things are slowly improving in recent years. As shown in this review, before 2010 there were only 7 cases of robotically treated VAAs published in the literature, whereas in the following decade (2011–2019) we found reports on 20 patients, and 27 more between 2020 and 2024: the use of robotic techniques in the treatment of VAAs is increasing at a rapid pace, although the only available data regarding its outcomes are limited to case reports and case series, at least for the time being.

The only systematic reviews published so far are those by Antoniou et al. (2011) and by Ossola et al. (2020) [26,27]. The first reports on the state of the art of robotic technology in vascular surgery, focusing on the experience on animal models; the latter compares the results of laparoscopic and robotic repair of SAAs, analyzing the results of 94 aneurysms treated laparoscopically and 13 treated by RALS [25]. The comparison showed no differences in terms of mortality, reinterventions, operative times, and length of in-hospital stay, but it reported better conversion rates after robotic surgery than laparoscopy (0 vs. 4.8%) [26].

Although such results are encouraging, the work published by Ossola et al. is limited by the small number of patients treated with robotic surgery in comparison with those treated laparoscopically. Moreover, the authors only cover splenic artery aneurysms, which represents just one possible application of robotic surgery to the wide world of visceral artery aneurysms, albeit the most common one. Finally, there is no comparison, so far, with the real competitors of RALS and laparoscopic treatment of VAAs, namely, endovascular and open surgery.

The topic is interesting and worthy of further studies because of the existing controversial aspects regarding the optimal elective treatment of VAAs, acknowledged by the most recent guidelines [5,6]. In fact, both open and endovascular options have well-known limits, and the studies available so far have failed to establish with an adequate level of evidence the recommended treatment for many types of visceral aneurysms.

The aim of our review is to define the initial success rates, mortality, morbidity, and long-term outcomes of RALS in all kinds of visceral artery aneurysms. To the best of our knowledge, there is no other systematic review of the advantages and limits of robotic surgery in the treatment of visceral arteries.

The current review reports on 30 cases of SAAs and 23 other VAAs, especially renal aneurysms, treated with robotic surgery, and it is an initial attempt to collect all the available information regarding RALS and VAAs.

Our results, collected four years after Ossola et al., confirm a remarkable increase in the employment of robotic surgery for VAAs, and they report extremely positive outcomes, although the small numbers do not allow any statistical comparison with open surgery and laparoscopic surgery in the same locations.

According to our results, one of the most important advantages of RALS is the extremely low rate of perioperative complications, thanks to minimal invasiveness. In fact, the data collected in this review show an absence of cardiac and respiratory complications, thromboembolic events, and infections, which are the most frequent events complicating open surgery of VAAs [4]. According to the metanalysis by Barrionuevo et al., the cumulative rate of such events range between 10 and 15% (cardiac adverse events prevalence: 1%, respiratory failure: 3%, surgical site infections: 3%, pulmonary embolism: 3%). Minor complications, such as bleeding and mild impairment of the liver and renal functions, were also rare and non-disabling for patients undergoing RALS for VAAs [5].

Possible disadvantages, instead, include the operative time and the periods of renal and splenic ischemia (30–40′). We do not have information based on large clinical trials concerning the operative and ischemia times of open surgery of VAAs and VAPAs, but it is possible that those of RALS are slightly longer [4,5]. That may be due to the learning curve and could be liable to improve in the future. However, according to the data reported here, such longer times did not cause significant adverse clinical outcomes.

Moreover, the robotic-assisted approach is characterized by early recovery of oral feeding and short in-hospital stay, especially following RAA repair. The reason for longer in-hospital stays after the repair of SAAs is not clearly understandable from the sources of this review, but it may be due to a more cautious approach related to the need for close monitoring of liver, spleen, and bowel vascularization, although the observed complications were not more frequent after SAA than RAA treatment.

The absence of such complications confirms the advantages of minimally invasive RALS, although the issue of early and late conversion is still a concern that should not be underestimated.

According to our data, the rates of intraoperative conversion to laparoscopic or open surgery (5.6%) are satisfactory but far from the 0% of the previous study, reflecting the higher number of cases analyzed here [26,27]. From the information we have collected, SAAs appear more prone to conversion than RAAs, and the most frequent cause of that is flat tissue adhesion, which hinders the optimal artery preparation and was the main reason for failure. This suggests that patients with a hostile abdomen may be at higher risk of failure and conversion, and correct patient selection is crucial to minimize such risk. There was also a case of failure due to a technical problem with the robotic arm, hindering the reconstruction of a splenic artery and causing a perioperative splenic infarction. The complications related to material failure are another aspect worthy of investigation, but for the time being, with such low numbers, it is impossible to estimate its risk and how it may affect the results of RALS.

All patients who are candidates for RALS should be made aware of the risk of conversion, like in the other minimally invasive options (endovascular and laparoscopic treatment), although such risk is still difficult to estimate due to the paucity of data. If the 5.6% conversion rate reported in this review is reliable, which must be confirmed by future research as higher-volume studies become available, it compares favorably with that of endovascular surgery, which is also minimally invasive but has the limitation of less-durable results (3% end-organ ischemia in RAAs and 15% in SAAs after 1 year) [5]. The low rate of end-organ ischemia (1.8%) and reintervention (1.8%) of robotic treatment in the mid-term (median follow-up: 9 months, range: 2–48) is another important advantage of this approach, although longer-term results are still needed for confirmation.

Reinterventions after RALS were due to splenic infarction, requiring partial splenectomy, or post-anastomotic renal artery stenosis, corrected by percutaneous angioplasty. Such complications, probably due to technical defects in anastomosis confection, can also be observed after traditional open surgery and might be related to the learning curve or technical mistakes. Notably, in the cases analyzed in this review, no post-anastomotic pseudoaneurysms or arteriovenous fistula were observed after robotic surgery, which suggests that the surgical precision of robots might make the arterial reconstruction more stable over time, as suggested by Tufano et al. [28].

Moreover, robotic repair, unlike endovascular exclusion, is not burdened by the risk of aneurysm reperfusion and rupture, which is the true Achilles heel of minimally invasive treatment.

A different, and unique, kind of robotic treatment of VAAs has been proposed by Aziz et al., who combined robotic technologies with endovascular treatment to embolize a renal artery aneurysm with hilar location and tortuous anatomy in a young female patient [23]. The authors reported improved navigability and precision of coil deployment in the renal artery, achieving selective embolization without sacrifice of the side branches. The procedure was successful, and the postoperative course uneventful until discharge, but follow-up is not reported. When applied to endovascular surgery, robotic technologies could be useful in facing anatomical and technical challenges, but it has no advantages in terms of durability, because the aneurysm is not resected, and it requires close surveillance with serial computed tomography angiography, just like standard endovascular repair. Therefore, it is our view that this new method should be avoided in young patients, who would benefit more from robotic-assisted laparoscopic surgery, and that endovascular treatment is reserved for elderly individuals.

In conclusion, RALS has several advantages, although its feasibility is still limited to selected cases. Unfortunately, there is no consensus or evidence guiding optimal patient selection, and the inclusion and exclusion criteria for candidate patients for RALS have not been defined yet. In fact, most authors mentioned in this review did not explain the reasons why robotic repair was preferred over the other options. However, based on our experience, which is limited to SAAs, we can state that robotic-assisted treatment has more advantages over endovascular treatment in aneurysms located at the splenic hilum, when endovascular embolization or endografts sacrifice collateral vessels leading to splenic infarction or post-embolization syndrome [12].

Although this is not discussed in the reports, it is possible that the same considerations could be extended to the renal district as well.

Based on our experience and on the RALS advantages reported in the literature, a robotic approach should be considered in the following cases:-Young patients who are at risk of reintervention in the longer term following endovascular repair.-Older patients or patients unfit for open repair with aneurysms with distal location or arising in proximity to a bifurcation of the parent artery, where endovascular exclusion would be more challenging and present a higher risk of failure.

Robotic treatment of VAAs, on the other hand, is at risk of open or laparoscopic conversion in case of a hostile abdomen, in which case an endovascular approach, either standard or robotic-assisted, should be preferred when feasible.

The present review has several limitations. The most important is the small sample size, which reflects the still rare employment of the robotic approach, limited by cost issues and surgeons’ experience.

The second is the extremely variable and generally short follow-up, which ranges between 2 and 48 months, with a median period of observation of 9 months. Late complications, such as anastomotic stenosis or pseudoaneurysms, might have been overlooked, and studies reporting on longer-term outcomes will be necessary. A publication bias is also possible considering the outstanding outcomes reported in terms of mortality and complications.

Moreover, the possible role of RALS in urgent repair is not discussed. Although emergent treatment in hemodynamically unstable patients is probably unfeasible with the robotic approach, RALS may be useful to treat contained ruptured VAAs, achieving a more durable result than endovascular surgery, especially in younger patients. This possibility has not been explored yet, but it will require some time to obtain more experience and confidence with elective treatment before RALS can also be tested in urgent settings.

Lastly, the results reported here allow some conclusions to be drawn, specifically regarding splenic and renal artery aneurysms, but their potential in treating other VAAs (coeliac trunk aneurysms, superior mesenteric aneurysms, colic, and pancreaticoduodenal aneurysms) remains largely unexplored.

## 5. Conclusions

Robotic-assisted repair of VAAs and VAPAs is an interesting therapeutic option that is less invasive than open surgery and more radical than endovascular exclusion. Although the outcomes must be further ascertained by studies on a larger population with a longer follow-up, the robotic technique is safe and effective in treating splenic and renal artery aneurysms and should be considered in all cases where endovascular and open treatment are at risk of technical failure or major complications.

## Figures and Tables

**Figure 1 jcm-13-03385-f001:**
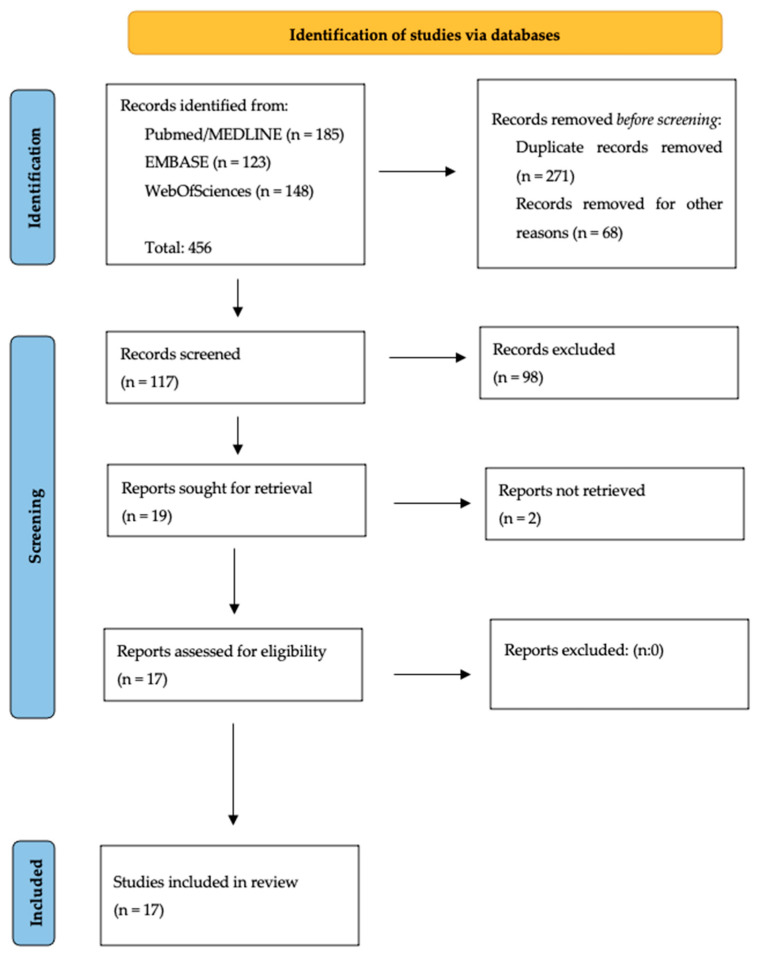
Selection of the studies according to the PRISMA Guidelines.

**Table 1 jcm-13-03385-t001:** Studies included in the review.

Authors	Pts (N)	Year	Location	Device	Selection Criteria	Technique
Luke et al. [22]	1	2006	RAA	NR	Aneurysm located at the renal hilum	Direct anastomosis
Pietrabissa et al. [24]	1	2010	SAA	NR	NR	Direct anastomosis
Giulianotti et al. [19]	5	2010	RAA	NR	NR	4 graft reconstruction, 1 direct anastomosis
Giulianotti et al. [21]	9	2011	SAA	NR	NR	7 direct anastomoses, 2 graft reconstrucion
Samarasekara et al. [20]	1	2013	RAA	DaVinci Si	Aneurysm located at branch point of renal artery	Direct anastomosis
Gheza et al. [25]	1	2013	RAA	NR	Aneurysm located at branch point of renal artery	Graft reconstruction
Salloum et al. [18]	1	2014	HAA	NR	Unsuitability for EVT for size and position	Excision and ligation
Stadler et al. [8]	4	2016	SAA	DaVinci Standard	NR	NR
Salloum et al. [17]	1	2016	CTA	DaVinci Si	Unsuitability for EVT for size and position	Excision and ligation
Long et al. [15]	1	2017	RAA	NR	Unsuitability for EVT (not specified)	Excision and repair
Wei et al. [16]	1	2017	RAA	NR	NR	Excision and repair
Aziz et al. [23]	1	2018	RAA	Magellan	NR	Coil embolization
Abreu et al. [14]	9	2020	RAA	DaVinci Si	NR	5 excision and repair, 3 direct anastomoses, 1 graft reconstruction.
Marone et al. [12]	4	2020	SAA	DaVinci Xi	Aneurysm located at the renal hilum	Direct anastomosis
Ceccarelli et al. [13]	1	2020	SAA	NR	Risk of PES and splenectomy	Direct anastomosis
Grandhome et al. [10]	1	2021	RAA	DaVinci Xi	Aneurysm located at branch point	Excision and patch repair
Wu et al. [11]	1	2022	RAA	DaVinci Xi	Aneurysm located at the renal hilum	Direct anastomosis
Stadler et al. [9]	11	2022	SAA	DaVinci Standard	NR	NR

SAA: splenic artery aneurysm, RAA: renal artery aneurysm, CTA: coeliac trunk aneurysm, HAA: hepatic artery aneurysm, EVT: endovascular treatment.

**Table 2 jcm-13-03385-t002:** Location, operative data, and follow-up of VAAs treated robotically.

Location	Pts (N)	Cross-Clamping Time (min)	Mean OT (Standard Deviation, SD)	Mean LOS (min)	Median F-U in Months (Range)	Conversions
**SAA**	30	45.5′ (SD: 0.5′)	130′ (SD: 50.2)	6 (SD:0.8)	12 (3–48)	3 (10%)
**RAA**	21	37.8′ (SD:10.7′)	218.4′ (SD: 66.4)	3.5 (SD:1.5)	16 (2–28)	0
**CT or HA**	2	NR	173.5	4.5	6	0
**Total**	**53**	**39.7′ (SD: 9.9′)**	**169.3′** **(SD: 83.3′)**	**4.2 (SD: 1.6)**	**9 (2–48)**	**3 (5.6%)**

OT: operative time, F-U: follow-up, SAA: splenic artery aneurysm, RAA: renal artery aneurysm, CT: coeliac trunk, HA: hepatic artery aneurysms.

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
