# Peer review of "Robotic Surgery for Elective Repair of Visceral and Renal Artery Aneurysms: A Systematic Review"

_jcm, 2024, doi:10.3390/jcm13123385_

Round 1

Reviewer 1 Report

Comments and Suggestions for Authors

TThis review is highly intriguing. Due to its pioneering nature, there are relatively few reported cases, and it is anticipated that there are more cases where the procedures were unsuccessful than those that have been reported, so there will be a need for the accumulation of many more cases from a statistical standpoint. On the other hand, there is sufficient value in this study as a review of pioneering studies in the field of robotic surgery in the vascular domain. Considering the potential for application not only in peripheral vascular surgery but also in aortic surgeries of the chest and abdomen, it appears to be a field with plenty of room for further development.

Author Response

We thank Reviewers 1 for his/her comment and we are glad for the appreciation expressed in this review. We agree that the numbers are too few to draw solid scientific conclusions about the value of RALS in visceral artery aneurysms and that there is a possible publication bias, as reported in the manuscript, but we think that data collection about this issue should be started somehow, and this is the goal of the article. 

With our best regards,

The authors.

Reviewer 2 Report

Comments and Suggestions for Authors

I read with great interest this review on Robotic surgery for elective repair of visceral and renal artery aneurysms. The idea of a review on this topic is certainly innovative. However, I have some concerns:

Introduction- definitely too brief. The authors should enhance this part.

Mat and methods: it seems more a systematic review than a narrative review. Authors have explored the major online databases as I can see from table 1. Authors correctly described the inclusion and exclusion criteria.

Discussion. authors should add a paragraph on the AVF after surgery, since its treatment is for sure a hot topic in surgery. Authors may refer to  doi: 10.1016/j.radcr.2022.04.038. 

This manuscript should be re-evaluated on a second round revision process.

Author Response

Dear Reviewer, 

Thank You for Your comments and Your help in improving our manuscript. Please find enclosed our answers with reference to the changes we have made according to Your suggestions. Please note that the indicated pages and lines refer to the redline manuscript.

1)Introduction: thank You for the invitation to expand this section. We have complied eagerly, by providing an overview about visceral artery aneurysms and the reasons why their treatment could benefit the most from endovascular treatment. (pages 1-2, lines 39-50: "Visceral artery aneurysms (VAA) and pseudoaneurysms (VAPA), despite their rarity in the general population, have a high impact in terms of mortality and morbidity due to the risk of sudden rupture, which is particularly threatening in pregnant women, patients affected by hepato-biliary diseases (portal hypertension, pancreatitis, transplant recipients), and in some forms of vasculitis [4,5]. The indications and the treatment modalities of intact aneurysms are still controversial, due to the paucity and heterogeneity of the available data and the complications of both open and endovascular repair [6]"

2) Thank you for the remark. We had started planning a narrative review, but then shifted to a systematic one, when we realized that the available data allowed for a systematic approach. As You suggested, we have modified the title and abstract accordingly,

3)Discussion: We are grateful to reviewer 2 for the suggestion to discuss the aspect of post-operative AVF: we have added this aspect in the discussion section, page 9, lines 328-331, with the proper citation ("Notably, in the cases analyzed in this review, no post-anastomotic pseudoaneurysms or arterio-venous fistula were observed after robotic surgery, which hints that the surgical precision of the Robot might make the arterial reconstruction more stable over time, as suggested by Tufano et al [28]").

Hoping that our changes will be useful to improve the manuscript quality and readability, we thank the reviewer and offer our best regards.

The authors

Reviewer 3 Report

Comments and Suggestions for Authors

This narrative review aims to collect the reported results of robotic-assisted surgery in the treatment of visceral artery aneurysms.

1In result section, authors stated that 13 cases (13 of 17 included) were justified, since those arterial aneurysm were the unsuitabile for endovascular exclusion due to extremely distal location. However, until now, the indications and inclusion and exclusion criteria for robotic aneurysm repair have not been clear.

2According to the results presented by the authors, we found that both RAA and SAA have a long operation and blocking time, especially for the kidney. Does such a long blocking time cause serious damage to the kidney function?In this case, is it reasonable to choose a robotic repair?

3In result section (the paragraph of peri-operative results), the author represent the result of SE difference of the time of ischemia, but not the mean time, please add this result to the article.

4In the discussion section, author stated that Laparoscopic and robotic treatment of VAAs is widely described in the literature and looked at with interest since it can combine the advantages of traditional surgery (radical treatment) and endovascular exclusion (minimal invasiveness). In fact, the treatment of interventional endovascular therapy is different from surgical treatment. Although Laparoscopic and robotic treatment can be minimally invasive, but it's not appropriate to state that robotic treatment combine the advantages of traditional surgery (radical treatment) and endovascular exclusion(minimal invasiveness).

5There are still some problems in the writing of this article, and some expressions are rusty.

Comments on the Quality of English Language

There are still some problems in the writing of this article, and some expressions are rusty.

Author Response

Dear reviewer, 

We are grateful to You and Your Colleagues for Your help in improving our manuscript and for Your comment. We have worked to satisfy at best Your requests and Your suggestions. Please, find enclosed our answers to Your queries. Please note that all pages and lines here indicated refer to the redline version of the re-submitted manuscript.

  1. We are grateful to the reviewer for raising the point of selection criteria. In fact, there is currently no indication about inclusion and exclusion criteria concerning RALS, and that is what would make it interesting to know how the authors of each publication chose their patients for this treatment. According to their experience and our own, we have drawn some conclusions about those cases that could benefit the most from RALS, and those which are more prone to complications or failure. We have reported as much in the results section (page 6, lines 176-177: "Although the current literature does not point out specific situations in which robotic surgery should be considered, selection criteria for RALS were specified in 13 cases"), and in the discussion section (page 10, lines 348-357: "In conclusion, RALS has several advantages, although its feasibility is still limited to selected cases. Unfortunately, there is no consensus or evidence guiding an optimal patient’s selection and the inclusion and exclusion criteria to candidate patients to RALS have not been defined yet. In fact, most authors analyzed in this review did not explain the reasons why robotic repair was preferred over the other options. However, based on our experience, limited to SAAs, we can state that robotic-assisted treatment has more advantages over endovascular treatment in aneurysms located at the splenic hilum, when endovascular embolization or endografts sacrifice collateral vessels leading to splenic infarction or post-embolization syndrome [12]").
  2. We agree with Reviewer 3 that long ischemia time could theoretically be a disadvantage of this treatment, although there were no clinical consequences according to our data.  We have discussed this matter at page 8, lines 285-290 ("Possible disadvantages, instead, include the operative time and the periods of renal and splenic ischemia (30-40’). We do not have information based on large clinical trials concerning the operative and ischemia times of open surgery of VAAs and VAPAs, but it is possible that those of RALS are slightly longer [4,5]. That may be due to the learning curve and could be liable to improvement in the future. However, according to the data here reported, such longer times did not cause significant adverse clinical outcomes.")
  3. We apologize for not having reported the mean ischemia times for SAAs and RAAs, which was 45.5’ and 37.8’ respectively, as now reported at page 6, line 197, and in table 2.
  4. We respect the reviewers view that it's not appropriate to state that robotic treatment combine the advantages of traditional and endovascular surgery, and we have rephrased this statement in a more conciliatory fashion ("Robotic-assisted repair of VAAs and VAPAs is an interesting therapeutic option, less invasive than open surgery, and more radical than endovascular exclusion), both in the abstract and in the conclusions (page 11, lines 391-394). We hope the reviewers will find this observation more appropriate.
  5. We are grateful for inviting us to revise the English, and we have submitted the text to a native speaker for a thorough language editing. All corrections and rephrasing are marked in the redline version throughout the text. We hope that we succeeded in taking the rust away.

With our best regards, 

The authors 

Round 2

Reviewer 2 Report

Comments and Suggestions for Authors

Authors addressed my major concerns and improved the quality of the manuscript

Author Response

We thank Reviewer #2 for his/her positive judgement, and we are glad that our manuscript can now be considered for publication. 

Best regards, 

The authors

Reviewer 3 Report

Comments and Suggestions for Authors

In Figure 2, the overall image quality is low, it is recommended to use other mapping software to improve the image quality and improve the structure to make the image look silky.

Author Response

We thank the reviewer for his/her work in revising our manuscript, and we appreciate his/her suggestions, but the manuscript does not contain any figure 2, and figure 1 is fine.